# Ideas and perspectives: Biogeochemistry – Some Key Foci for the Future

Thomas S. Bianchi[1], Madhur Anand[2], Chris T. Bauch[3], Donald E. Canfield[4], Luc De Meester[5,6,7], Katja Fennel[8], Peter M. Groffman[9], Michael L. Pace[10], Mak Saito[11], Myrna J. Simpson[12]

[1]Dept. of Geological Science, University of Florida, Gainesville, FL USA
[2]School of Environmental Sciences, University of Guelph, Guelph, Ontario, Canada
[3]University of Waterloo, Department of Applied Mathematics, Waterloo, Canada
[4]Nordcee, University of Southern Denmark, Odense, Denmark
[5]Dept. of Biology, University of Leuven, Leuven, Belgium
[6]Leibniz Institut für Gewässerökologie und Binnenfischerei (IGB), Berlin, Germany
[7]Institute of Biology, Freie Universität Berlin, Berlin, Germany
[8]Dept. of Oceanography, Dalhousie University, Halifax, Nova Scotia, Canada
[9]City University of New York Advanced Science Research Center at the Graduate Center, New York, NY USA and Cary Institute of Ecosystem Studies, Millbrook, NY USA
[10]Dept. of Environmental Sciences, University of Virginia, Charlottesville, VA USA
[11]Marine Chemistry and Geochemistry, Woods Hole Oceanographic Institution, Woods Hole, MA USA
[12]Dept. of Physical and Environmental Sciences, University of Toronto, Toronto, Canada

*Correspondence to*: Thomas S. Bianchi (tbianchi@ufl.edu)

**Abstract.** Biogeochemistry has an important role to play in many environmental issues of current concern related to global change and air, water, and soil quality. However, reliable predictions and tangible implementation of solutions, offered by biogeochemistry, will need further integration of disciplines. Here, we refocus on how further developing and strengthening ties between biology, geology, chemistry, and social sciences will advance biogeochemistry through: 1) better incorporation of mechanisms, including contemporary evolutionary adaptation, to predict changing biogeochemical cycles; and 2) implementing new and developing insights from social sciences to better understand how sustainable and equitable responses by society are achieved. The challenges for biogeochemists in the 21st century are formidable and will require both the capacity to respond fast to pressing issues (e.g., catastrophic weather events and pandemics) and for intense collaboration with government officials, the public, and internationally funded programs. Keys to success will be the degree to which biogeochemistry can make biogeochemical knowledge more available to policy makers and educators about predicting future changes in the biosphere, on time scales from seasons to centuries, in response to climate change and other anthropogenic impacts. Biogeochemistry also has a place in facilitating sustainable and equitable responses by society.

## 1. Introduction

Biogeochemistry was one of the first truly inter- or multi-disciplinary sciences (Bianchi, 2020; Gorham, 1991; Schlesinger, 1991; Vernadsky et al., 1926) and the field continues to expand in multiple directions at an amazing pace; from small scales via interactions with microbiology and omics approaches (genomics, transcriptomics, proteomics, and metabolomics) (Figure 1) to large scales as a component of Earth System Sciences (Steffen et al., 2020).

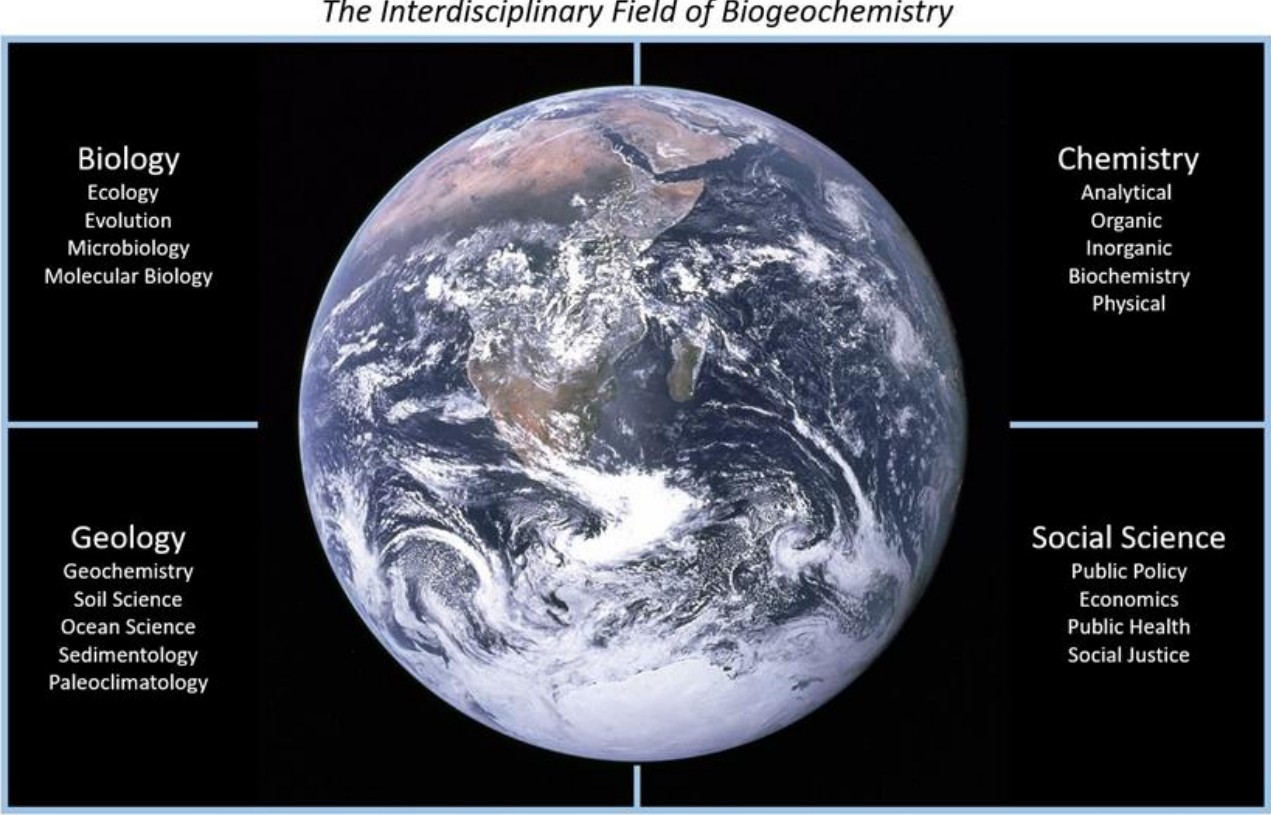

**Figure 1. The interdisciplinary field of biogeochemistry has advanced with expanded approaches that involve the integration of core biogeochemical areas with other disciplines needed to understand a rapidly changing Earth system and meet the needs for sustainability.**

A recent review of Biogeochemistry by Bianchi (2020), reflects on these more nascent linkages in molecular biology and their historical and disparate connections. Countries around the world scramble with new health and sociopolitical challenges ranging from global climate change, to loss of biodiversity, changing ecosystems, and global pandemics (e.g., coronavirus disease 2019 [COVID-19]) - often linked with the growing human population. Addressing these challenges requires new cross-disciplinary approaches by scientists - coupled with a better science–educated public that are more involved with decision-

making on sustainability issues (e.g., Derouin, 2020). Some may argue these are "old" questions/challenges previously debated in the academic circles of Earth Science, Ecology, Ecosystem Science, Biogeochemistry, and the like. However, we contend

that humanity now sits at a "Silent-Spring Moment" with the attention of the global public – in large part due to COVID-19 and recent catastrophic weather events. Recognition of earlier "moments" of environmental crisis led to the formation of organizations like the World Climate Research Programme (WCRP), International Geosphere-Biosphere Programme [IGBP]) and the Intergovernmental Panel on Climate Change (IPCC), to name a few. Kress et al. (2020) posit that "Now is the time to use the full power of science through cooperative efforts among initiatives such as BIOSCAN, the Global Virome Project

(GVP), and Earth BioGenome Project (EBPP) to advance our understanding of the complex web of interactions that span the domains of life." So, how can biogeochemists assist in linking human population dynamics, range expansion effects on element cycling, organismal adaptation, and contaminant cycling, to name a few, with such global efforts? It has long been recognized that better integration of environmental sciences and social sciences are needed in seeking a viable sustainability for the future. For example, the introduction of *Translational Ecology* (Schlesinger, 2010), as well as continued emphasis by

the National Science Foundation (NSF) and their supported working groups, such as the National Socio-Environmental Synthesis Center (SESYNC [https://www.sesync.org/]), clearly reflect new solutions to the problem.

Are biogeochemists adequately unified in addressing some of these key global issues in the 21st Century? Most biogeochemists would agree that better links are needed with Earth System Models, including better links between biogeochemical cycling,

organismal traits and their changes, and environmental modeling. This is a major challenge requiring connections between cellular and organismal level systems biology with observational and modeling studies of global biogeochemical cycles. Synergies between detailed process-level understanding through local or regional studies, and the ability to upscale and detect global change through global-scale observations, have already contributed strongly to progress in our field, advancing beyond some its previously conceived shortfalls (Cutter, 2005; Likens, 2004). Nevertheless, biogeochemists, amongst others, have

called for more improvements (Groffman et al., 2017) in the accessibility and sharing of complex data (Saito et al., 2020a; Tanhua et al., 2019; Villar et al., 2018), the integration of observations and predictive models (Fennel et al., 2019), and the incorporation of societal factors (e.g., damming, nutrient management) in model projections (Seitzinger et al., 2010). Here, we call for better incorporation of mechanistic knowledge from "omic" studies (Urban et al., 2016; Coles et al., 2017) and a stronger integration of modern and past ecological and evolutionary dynamics with biogeochemistry. Climate-driven range

expansion of organisms, including immigration-emigration patterns by humans, are expected to enhance zoonotic diseases (both viral and bacterial) (e.g., Han et al., 2015; Allen et al., 2017) and threaten global food supplies (via rise in soil pathogens) (e.g., Delgado-Baquerizo et al., 2020). These changes are likely to be coupled with broader shifts in community-level interactions, organismal adaptive change (Scheffers et al., 2016), and associated changes in biogeochemical cycling rates and fluxes (e.g., nutrient, contaminants, redox conditions etc.) (e.g., Bianchi et al., 2021). In this perspective, we provide some

insights on why these linkages, between biogeochemistry, evolutionary biology, and social sciences, are what we believe to be some key foci for biogeochemists in the coming decades. Thus, we argue for 1) better integration of adaptive evolutionary

change, coupled with range expansion, and biogeochemical cycles, and 2) continued integration of social sciences, focusing on the human-natural system - in the context of sustainability and biogeochemistry.

## 2. Eco-Evolutionary Dynamics and Biogeochemistry

There has been a longstanding interest in the co-evolution of life and biogeochemical cycles on Earth, as chemical conditions of this planet have been strongly influenced by evolving biochemical capabilities of life (Canfield et al., 2007; Lenton et al., 2014; Saito et al., 2003). Earth's life support system is inextricably tied to biogeochemical cycling and prokaryote evolution (Falkowski et al., 2008). Research has shown that understanding such co-evolutionary patterns in prokaryotes are key in developing environmental engineering solutions for future sustainability in the Anthropocene (Newman and Banfield, 2002).

Also, metazoans have long been recognized as important "engineers' of Earth's elemental cycles (Darwin, 1881). Moreover, a better understanding of organismal/biogeochemical interactions in the fossil record can help predict future linkages between climate change, organismal adaptation, range expansion, and biogeochemical cycles (Bianchi et al., 2021). For example, range expansion and/or contraction of marine benthic communities may be linked to climate change (Buatois et al., 2020). Furthermore, evolutionary radiations in marine environments reflect changing feeding guilds and bioturbation activities

(Mángano and Buatois, 2014), which show important biogeochemical feedbacks (e.g., redox changes in sediments) (Boyle et al., 2014; Buatois et al., 2020). We argue that more structured collaboration (via joint workshops and meetings) between ecological, biogeochemical, evolutionary, and paleoenvironmental scientists is needed in the 21st century to better utilize the fossil record for such questions.

Recently, evidence has shown that significant evolutionary trait change can occur over time scales of just a few generations,

and the rapidly changing environmental context at local, regional, and global scales during the Anthropocene, leads to strong selection pressures on populations to adapt (Bell and Collins, 2008; Hutchins et al., 2015; Kuebbing et al., 2018; Seibel and Deutsch, 2020). In fact, there is now ample evidence for rapid evolution, where rapid refers to contemporary evolution or evolution in ecological time. Multiple review papers and books on this topic have emerged (Bell and Collins, 2008; Fussmann et al., 2007; Hutchins et al., 2019; Palumbi, 2002; Schoener, 2011), as well as a monograph on Eco-evolutionary Dynamics

(Hendry and Green, 2017). In brief, from microbes to plankton, insects to plants, fishes, and birds, there are now hundreds of studies showing significant evolutionary change in trait values over short time spans - just a few tens of generations. Interestingly, this contemporary evolution impacts ecosystem functioning and elemental cycling dynamics (Bassar et al., 2010; Declerck et al., 2015). In one case, the evolution of zooplankton within a single growth season has been shown to shape the typical seasonal dynamics of phyto- and zooplankton in lakes (Schaffner et al., 2019). In another example, evolution in body

size in salmon, through its effects on salmon consumption by bears, impacts nutrient transfer from aquatic to terrestrial systems (Carlson et al., 2011). Many additional surprising pathways and mechanisms, inclusive of ecological aspects like behavior, remain to be discovered. For example, fear of predation by spiders can alter the elemental composition of grasshoppers, resulting in changes in production and nutrient cycles in ecosystems (Hawlena et al., 2012). Theory indicates that rapid

evolutionary trait change can also influence the occurrence and (recovery) trajectory of ecosystem regime shifts (Dakos et al., 2019). Integrating ecological and evolutionary responses is needed to make reliable predictions of how ecosystems respond to climate change (Matthews et al., 2011) and how this impacts biogeochemical cycles. While the evolutionary biologists, are well aware of such changes, better linkages with ecosystem ecologists and biogeochemists are needed to examine how these adaptive changes in taxa and community composition impact biogeochemical cycles – and how this gets integrated in future IPCC reports.

Rapid evolutionary changes are gaining attention because they can influence ecological responses, including responses to global change where new steep/novel gradients in "critical zones," continue to develop in the Anthropocene - in both terrestrial and aquatic systems (e.g., Bianchi and Morrison, 2018; Chorover et al., 2007). This implies that our analyses of ecological responses and their biogeochemical implications should not assume that trait values of species are fixed in time, and in equilibrium with biogeochemical rates and fluxes. Depending on the taxon and the selection pressure, traits can significantly change, and these changes have been shown to influence ecosystem processes such as consumption, production, respiration and nutrient cycles. Given the importance of microbes for biogeochemical cycles, this notion becomes even more important, because microbial generation times are short and can evolve significantly different trait values in a matter of a few days or weeks. For example, Lawrence et al. (2012) showed that when competing bacterial strains where forced to grow together, they changed their physiology so much that they became partially dependent on each other, and reached higher densities than when initially grown in the presence of one other. Yet, these observations of rapid evolution must be reconciled with the fact that the evolution of novel biochemical pathways, and their impacts on biogeochemical cycles, have occurred rarely (David and Alm, 2011). Eco-evo dynamics are generally underexplored relative to biogeochemistry thus, we implore this topic is a research frontier where integration among disciplines will led to significant advances in the 21st century.

In recent decades, the ability to directly track genes and gene functions of organisms in nature, especially in microbes, has greatly contributed to understanding their interactions with biogeochemical cycles (Martiny et al., 2006: Rusch et al., 2010). Measurements of microbial transcripts and proteins in natural environments has allowed direct observation of cellular functions as adaptive responses to the environment (Bergauer et al., 2018; Gifford et al., 2011). These functional systems include biogeochemically-relevant enzymes, transporters, storage molecules, and regulatory systems, and the quantitation of enzymes can be used to generate 'omic-based potential biogeochemical rates (Saito et al., 2020b). This provides mechanistic information about the underpinnings of biological controls on biogeochemistry and allows direct quantification of rate changes along different pathways. There are, however, many knowledge gaps to fill: roughly half of all genes have unknown function, the systems biology controlling gene regulation is poorly characterized (Held et al., 2019), and we know little about how the different biochemical pathways relate to resilience at the ecosystem level. Forging connections between the genetic and biochemical underpinnings to the production of metabolites that contribute to carbon and other element cycling is primed for discovery (Soule et al., 2015). Omics studies will also help reveal how the microbiomes of plants, invertebrates and

vertebrates, who make up the predator-prey and decomposition food webs, influence biogeochemical cycles (Macke et al., 2017).

**3. Embracing the Social Dimension**

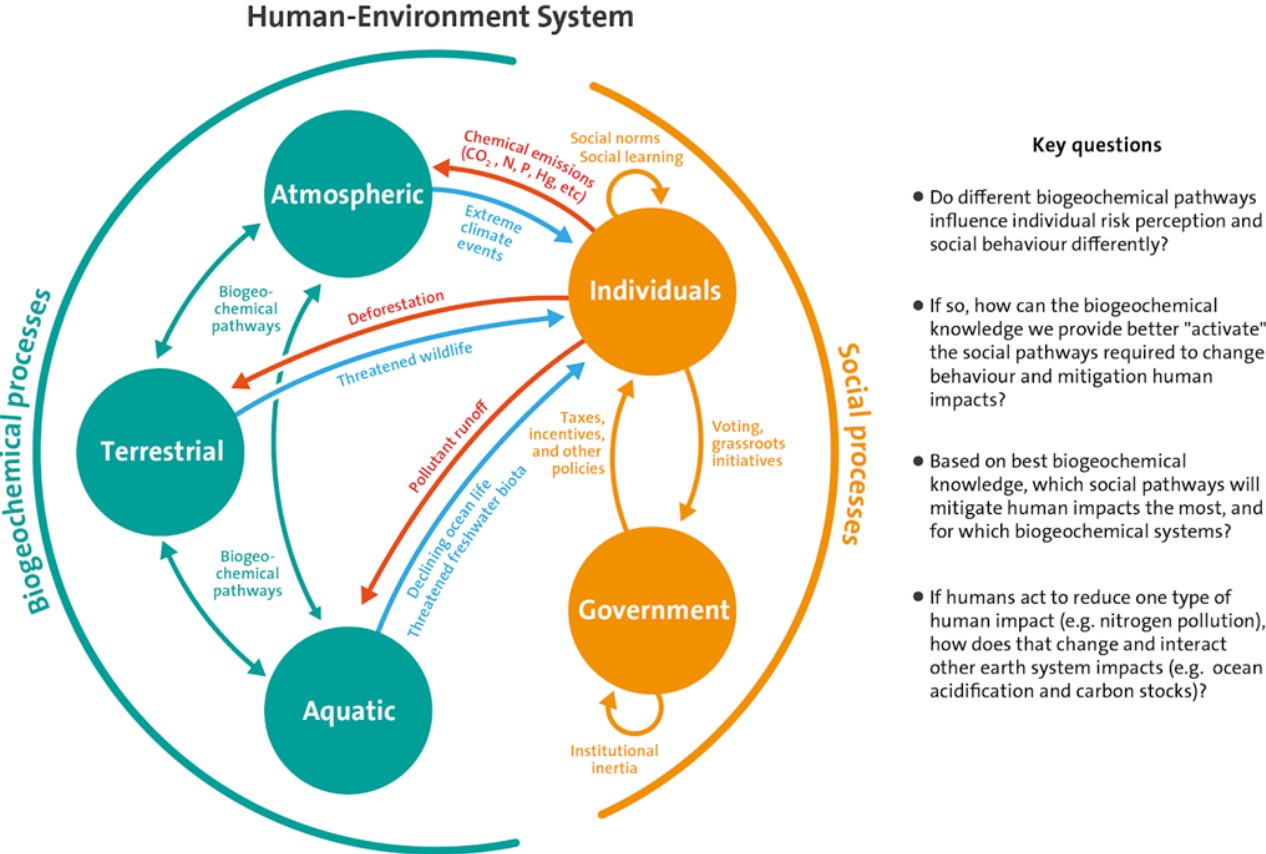

**Figure 2. "Translational biogeochemistry" would encompass the two-way feedbacks between human and biogeochemical systems. Understanding human societal pathways will be critical in discerning and mitigating future climate patterns and its biogeochemical consequences.**

With growing needs to understand how human population growth will interact with ecosystem evolution, in the face of climate change and organismal range expansion (including humans), biogeochemistry should continue to explore new dimensions of sustainability throughout the research process - including linkages with the social sciences for a holistic understanding of the human-environment system (Figure 2).   These ideas build on earlier notable efforts such as *Translational Ecology* (Schlesinger, 2010) and are similar to the idea of translational medicine, an interdisciplinary area of research that aims to

improve human health by accelerating the application of novel research discoveries to improving patient outcomes. Other fields, such as economics, are also warming to mission-driven research (Crain et al., 2014; Mazzucato, 2018). We argue that there is a large gap between discoveries in biogeochemistry and their application to improving ecosystem health. For example, biogeochemistry research may present many possible solutions to managing the global carbon budget - from planting a trillion trees, to carbon taxes and trading, to direct air carbon capture - but their adoption (or not) by various levels of society are not generally studied scientifically, which could very much hamper the development of solutions.

As human range expansion continues to escalate with climate-change driven food security issues in many countries, particularly developing nations (Carney and Krause, 2020) integrating human behaviour at several scales is essential.  In some cases, a lack of understanding of social processes can lead to unexpected societal "push back", rendering scientific knowledge less impactful.  In 2018, the "yellow vest" protests in France emerged in response to a new carbon tax and greatly hindered progress toward carbon management goals in that country.  Protestors agreed that climate change is an issue but were not willing to accept socially unjust solutions.  Top-down approaches like the Paris agreement, with its ongoing political challenges and limited efficacy in combating climate, face severe challenges, therefore scientists are rushing to study how to harness social forces in a polycentric manner in order to tackle global-scale sustainability challenges.  Put succinctly, without involving individuals outside of the field in all stages of the research process, biogeochemistry research that seeks to advance sustainability through policy or behaviour change risks answering questions that decision-makers are not asking, or proposing solutions that populations will not adopt.  The window of opportunity to protect many of the natural systems we have the privilege of studying is rapidly disappearing.  And, the primary barriers to adoption of many sustainability solutions are often political and social limitations, not lack of scientific knowledge or availability of technology.  Increasing public awareness of how basic science is linked with environmental problems through early education will be key in reducing these limitations.

Does the need for socially conscious policy-driven research mean that basic science inquiry in biogeochemistry is dead? Are we not allowed to wonder about the origin of the earth and the basic processes that underlie the cycling of energy, water, and nutrients across the surface of the earth? We argue that there is a false dichotomy between biogeochemistry and a translational bio-geo-socio-chemistry. Indeed, our goals are consistent with Vernadsky's original vision, as he stated "understanding our planet the way it is." The difference is that human influence on the Earth system is now so pervasive that our challenge has moved from integration of biology, geology and chemistry, to inclusion of social sciences where evidence now exists for how organismal range expansion and/or contraction of say, marine benthic communities respond to climate change (Buatois et al., 2020). Indeed, by including the social sciences, biogeochemistry can help predict future changes in the biosphere in response to climate change and other anthropogenic pressures, while helping to facilitate sustainable and equitable responses by society. Thus, biogeochemical knowledge comes closer to policy makers. Nevertheless, our knowledge of biogeochemical cycles remains relatively limited compared to other core sciences (biology, chemistry, physics, geology) and thus basic research will be key in understanding and laying the foundation for good policy development - relating to global change

science. While the core scientific disciplines of physics, chemistry, geology, and biology, have long been essential in the development in part, of NASA exploration, pharmaceutical and engineering materials, petrochemical processes, and medicinal and agricultural biochemistry, respectively, biogeochemistry is uniquely poised (see Figure 1) to serve the public as a key core science in addressing climate change issues.

Biogeochemistry is already an inherent component of Sustainability and Earth System Sciences, which are addressing the overarching challenge of how global change pressures the habitability of the planet and the ability to sustainably use its resources to feed and supply the world population and economy. A pivotal issue is how organismal, environmental, and societal processes cause feedbacks that affect biogeochemical cycles and global change (Seitzinger et al., 2010). A "translational" biogeochemistry would be a natural pathway of research on transformational human-environment processes because: (1) both sustainability science and biogeochemistry are systems science approaches, and (2) collaboration between biogeochemists and social scientists could address topical key questions at a scale that is both holistic with respect to social-climate interactions, and suitably detailed in addressing biogeochemical issues (Figure 2). A holistic human-environment systems approach to applied biogeochemistry that accounts for social feedback might help winnow down policy recommendations to those that are both effective and likely to be adopted. Calls for greater integration between social and natural sciences have been made for years. The barriers slowing this integration are not only due to differences in methodology and perspectives. We speculate that lack of data on the interactions between social and natural systems has also contributed to this problem. However, we suggest that the dawn of digital social data has created a vast amount of essentially free observational data on social systems and their relation to natural systems that could help address this limitation and should be taken advantage of.

Key questions include: Do changes in biogeochemical pathways, associated with specific climate-change drivers (e.g., droughts/flooding events versus ocean acidification), influence risk perception and social behaviour differently than other broader global change issues (e.g., GHG emission reduction versus dietary change)? If so, how can the biogeochemical knowledge we provide better "activate" the social pathways required to support mitigation behaviour? And, how will feedbacks between different biogeochemical systems hinder or accelerate these social pathways? An example of such an integrative approach is coupled social-climate modelling (Bury et al., 2019), in which sub-models are developed both for social dynamics and climate dynamics, and the two sub-models are then coupled together. As a result of this, socio-economic pathways become a prediction of theoretical models and thus become the subject of scientific study themselves, instead of being assumptions that are simply input into climate models. A human-environment perspective would change not only how we think about the natural world, but how we design our research. A sustainability science approach to research may include stakeholders and policy experts at all stages in the research process.

## 4 Summary

The regional and global importance of biogeochemical processes for the homeostasis of Earth's life support system necessitates accelerating research to achieve the goal of a sustainable global society. Starting from an awareness of the field's history, new developments and key limitations of current approaches, we aimed to develop a perspective on how biogeochemistry can better serve society.

The challenges and opportunities of 21st century biogeochemists are formidable and will require intense collaboration with government officials, the public, internationally-funded programs, and other fields in the social sciences. A key to success will be the degree to which biogeochemistry succeeds in making biogeochemical knowledge more available to policy makers and educators, predicting future changes in the biosphere in response to climate change (and other anthropogenic impacts on time scales from seasons to centuries), and in facilitating sustainable and equitable responses by society. Biogeochemistry can have an important role in bringing about a sustainable future. But, there are several impediments to fully realizing this role, including the need for further integration across disciplines and spatial scales, the intrinsic challenges of combining increased breadth with mechanistic depth, and the need to strengthen connections to society. While biogeochemistry has made major achievements in the past century describing Earth's global and regional biogeochemical cycles for the first time, we recognize that the field has acquired new societal responsibilities, in particular uncovering how humans are rapidly changing biogeochemical cycles, assessing the impact of these changes on biological communities and feedbacks on society, and effectively communicating this information to policy makers and society-at-large.

We call for more focused cross-disciplinary workshops and meetings, in an already complex mixture of interdisciplinary sciences, that allows for biogeochemistry (both paleo and modern) to utilize its novel origin and evolution (Bianchi. 2020), to develop a better way forward in planning for climate change. Perhaps a global congress of biogeochemical/climate change is needed to move this ahead.

*Competing interests:* The authors declare that they have no conflict of interest.

*Acknowledgements:* All authors contributed equally to the conceptualization and writing of this document. was

*Financial support:* T.S.B. was supported in part, by the Beverly Thompson Endowed Chair in Geological Sciences; M.J.S. acknowledges support from a Natural Sciences and Engineering Research Council of Canada Tier 1 Canada Research Chair in Integrative Molecular Biogeochemistry

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
