# Peer review of "Ideas and perspectives: Biogeochemistry – Some Key Foci for the Future"

_Biogeosciences, 2020_

## Referee Comment (RC1) · William Schlesinger (Referee) · 12 Nov 2020

I have mixed reaction to this paper. On the one hand, how can one criticize a plea for more interdisciplinary perspectives, especially into genomics and the social sciences? And one certainly cannot criticize pleas for better communication of our results to improve science-based policy. On the other hand, all this has been said before, widely and frequently, so I really didn't learn much from this perspectives piece, nor did it offer new excitement. The very name: Bio-Geo-Chemistry, reflects the interdisciplinary roots of this field of endeavor. In several papers, Paul Falkowski and Diane Newman have reviewed how the evolutionary biochemistry of prokaryotes has left its mark on

the Earth's chemical conditions. Even the importance of integrating the social sciences in environmental science is widely recognized—witness a program of coupled human and natural systems at NSF and a requirement by the agency for social science linkages in its long-term ecological research programs. Better transmission of our results to policy makers was named Translational Ecology by Schlesinger in a 2010 editorial in Science and now the focus of a working group at the National Socio-Environmental Synthesis Center (SESYNC) at the University of Maryland. I have a fear that the statements about rapid evolution (page 3, line 79ff) are at odds with the recognized inefficiency of eugenics. So, I have no real criticisms of this piece; I just question its novelty and potential impact amongst the crowded pages on our computers that already sap our time and energy for forward progress.

―――――――――――――――――――

---

## Author Comment (AC1) · 16 Nov 2020

Response to:Reviewer comment: I have a fear that the statements about rapid evolution (page 3, line 79ff) are at odds with the recognized inefficiency of eugenics.

Apart from our surprise with respect to the link to eugenetics that the referee makes, we feel that this comment strengthens us in the fact that we touch upon a novel perspective that needs to be highlighted in order to push the further development of the field of biogeochemistry. There is now ample evidence for so-called "rapid" evolution or, "rapid" here meaning: contemporary evolution or evolution in ecological time. Multiple review papers on this topic have emerged (Fussmann et al 2007 Functional Ecol-

ogy; Schoener 2011 Science,…), as well as a first monograph (Hendry, 2017, "Eco-evolutionary Dynamics", Princeton University Press). In brief, from micro-organisms over plankton and insects to plants, fish and birds, there are now hundreds of studies showing significant evolutionary changes in trait values over short time spans – just a few or a few tens of generations, leading to trait change during the course of a few weeks, a few months, or a few years. These evolutionary changes are rapidly gaining attention because they can influence ecological responses to, amongst others, global change. This implies that in our analyses of ecological responses and their biogeochemical implications, we should not assume that trait values of species are fixed in time. Depending on the taxon and the selection pressure, traits can significantly change, and these changes have been shown to influence ecosystem processes such as consumption, production, respiration and nutrient cycles. Given the importance of microbial organisms for biogeochemical cycles, this notion becomes even more important, because microbial organisms because of their short generation times can evolve significant different trait values in a matter of a few days or weeks. Just to illustrate with one example: Lawrence et al (2012) showed that bacterial strains that were competing with each other and had difficulties to grow together, when forced to grow together changed their physiology so much that they started to be partially dependent on each other, and thus reach higher densities when grown in the present of the other species. There are many ramifications through which eco-evolutionary dynamics can influence biogochemical cycles and recommendations, yet few people in the field are aware of that. So we view this as one of the next frontiers on which biogeochemistry can and should become even more integrative than in the past. We acknowledge that perhaps we were not clear enough on this point, so we edited the text in the revision we made.

---

## Referee Comment (RC2) · Anonymous Referee #2 · 22 Dec 2020

This paper has an excellent group of authors. However, I agree with the comments of Bill Schlesinger – it's not clear what hasn't already been said many times before. The abstract drives that home. What is really new in this paper?

So what might be a way forward? One suggestion is to develop a list of what are the new insights in this manuscript, and write the text around those instead of trying to cover so many different angles (scales, topics, etc).

That said, the paper is well written in that it was very easy to read. I just didn't come away with new insights.

---

## Author Comment (AC2) · 13 Jan 2021

Bianchi et al.: Response to Reviewers: We appreciate the suggestions from both reviewers and outline here a path forward that we believe will improve the paper. In particular, we acknowledge that not all of the issues raised in our paper are novel, and this is not what we had intended. Perhaps the main objectives of this exercise, which we carefully thought about through revisions and discussions, have been misconstrued by our choice of the title of the paper (Biogeochemistry: Its Future Role in Interdisciplinary Frontiers). We would like to change the title to be more reflective of what we consider to be a call to recharge and/or reassess of trends in biogeochemistry that are

ongoing, rapidly developing, and/or in need of greater emphasis over the next decade. Nevertheless, there are indeed a few novel questions raised in our manuscript which we can clarify in this response. In essence, our revisions would entail the following: 1) a new title for the paper with some further clarification of our main goal; and 2) more clearly emphasizing the key points (discussed below) - with additional references. Rev. 1. 1. "I have mixed reaction to this paper. On the one hand, how can one criticize a plea for more interdisciplinary perspectives, especially into genomics and the social sciences? And one certainly cannot criticize pleas for better communication of our results to improve science-based policy. On the other hand, all this has been said before, widely and frequently, so I really didn't learn much from this perspectives piece, nor did it offer new excitement." The issues raised in this paper were not meant to be novel, but more a reminder of where biogeochemistry needs to keep moving and where enhanced development and greater efforts are needed, which some specific examples of where past efforts have not yet succeeded. The audience is also intended to be broad including not only established biogeochemists but also early career scientists new to the field who may be particularly interested in its societal relevance. As we mention in our perspective, the Biogeosciences were only marginally prepared to understand how changes in economic activity associated with COVID19 affected air and water quality and carbon fluxes. Similarly, the necessary multidisciplinary interactions are still not well represented in new funding programs in such areas as Critical Zone science and Biological Integration in the U.S. and elsewhere. These along with other key topics in our paper are reminders of what this team of biogeochemists' view as key areas and something we feel is worthy of publication. 2. "The very name: Bio-Geo-Chemistry, reflects the interdisciplinary roots of this field of endeavor. In several papers, Paul Falkowski and Diane Newman have reviewed how the evolutionary biochemistry of prokaryotes has left its mark on the Earth's chemical conditions." Yes, and the paper by Falkowski, which is a key figure in a recent paper published by Bianchi (2020), on the history of biogeochemistry, along with work by Newman, would be good to mention as basis from which to continue building linkages between Earths' history

and future. This is one of the key points we emphasize in the paper, that is the importance of using proxies and fossil communities to better understand not just changes in community structure, but the impact these metazoan (not just microbial) changes have had, and will have, on key biogeochemical drivers, such as redox. While range expansion has been a notable topic in recent years, much of the emphasis has been on the consequences of changing diversity and microevolution, and not how these changes affect community structure along with associated biogeochemical properties and processes, for example, redox and changes in bioturbation and/or bio-erosion. We do discuss this in the article but will make sure this key point is clearly illustrated. 3. "Even the importance of integrating the social sciences in environmental science is widely recognized program of coupled human ËĞ and natural systems at NSF and a requirement by the agency for social science linkages in its long-term ecological research programs. Better transmission of our results to policy makers was named Translational Ecology by Schlesinger in a 2010 editorial in Science and now the focus of a working group at the National Socio-Environmental Synthesis Center (SESYNC) at the University of Maryland." While these points about multidisciplinary interactions, especially with social science, have been made before, little progress has actually been made to date, and a reassertion these issues along with other newly developing complexities are needed. For example, while translational biogeochemistry was certainly a notable and timely concept, something we clearly missed and will certainly cite in our revisions, we have witnessed recently protests related to carbon taxes that indicate need for new approaches and integration of biogeochemistry in relation to social science, and in particular the feedbacks with human behavior. The latter is a very embryonic line of research which we feel may be new to many in the scientific community. Once again, a reassessment and unification of ideas in our paper looking ahead for the next decade or so seems important, especially with broad the diversity of fields identify themselves under the Biogeosciences "umbrella". 4. "I have a fear that the statements about rapid evolution (page 3, line 79ff) are at odds with the recognized inefficiency of eugenics. So, I have no real criticisms of this piece; I just question its novelty and potential impact

amongst the crowded pages on our computers that already sap our time and energy for forward progress." As we have already responded to this comment (see below), we now acknowledge that perhaps we were not clear enough on this point, and will make the necessary revisions - as space permits. There is now ample evidence for rapid evolution, where rapid refers to contemporary evolution or evolution in ecological time. Multiple review papers on this topic have emerged (Collins and Bell, 2006; Fussmann et al., 2007; Hutchins et al., 2019; Schoener, 2011), as well as a first monograph (Hendry, 2017, "Ecoevolutionary Dynamics", Princeton University Press). In brief, from micro-organisms to plankton, insects to plants, fishes, and birds, there are now hundreds of studies showing significant evolutionary changes in trait values over short time spans –just a few or a few tens of generations, leading to trait change during the course of a few weeks, a few months, or a few years. These evolutionary changes are rapidly gaining attention because they can influence ecological responses to, amongst others, global change. This implies that in our analyses of ecological responses and their biogeochemical implications, we should not assume that trait values of species are fixed in time. Depending on the taxon and the selection pressure, traits can significantly change, and these changes have been shown to influence ecosystem processes such as consumption, production, respiration and nutrient cycles. Given the importance of microbial organisms for biogeochemical cycles, this notion becomes even more important, because microbial organisms because of their short generation times can evolve significant different trait values in a matter of a few days or weeks. Just to illustrate with one example: Lawrence et al (2012) showed that bacterial strains that were competing with each other and had difficulties to grow together, when forced to grow together changed their physiology so much that they started to be partially dependent on each other, and thus reach higher densities when grown in the present of the other species. There are many ramifications through which eco-evolutionary dynamics can influence biogeochemical cycles and recommendations, yet few people in the field are aware of that. So, we view this as one of the next frontiers on which biogeochemistry can and should become even more integrative than in the past. Rev. 2. 1. "This paper has

an excellent group of authors. However, I agree with the comments of Bill Schlesinger – it's not clear what hasn't already been said many times before. The abstract drives that home. What is really new in this paper?" See our aforementioned response to Reviewer 1 on this issue of novelty. 2. "So what might be a way forward? One suggestion is to develop a list of what are the new insights in this manuscript, and write the text around those instead of trying to cover so many different angles (scales, topics, etc). That said, the paper is well written in that it was very easy to read. I just didn't come away with new insights." We plan to make sure our points are more clearly listed. The reviewer, however, thinks the issues we discuss are valid and significant. We plan to revise the paper to enhance clarity by a change in the title and a more specific listing of our major topics. Overall, our view is the perspective we offer provides value for the biogeochemistry community through organizing and assessing key areas for progress. We intend that our essay stimulates assessment. We advocate for advances in areas we currently recognize, and we anticipate that via discussion and debate differing perspectives held by others will emerge to drive the field forward.

―――――――――――――――――――――――

---

## Author Response (AR1)

**Bianchi et al.: Response to Reviewers:**

We appreciate the suggestions from both reviewers and outline here a path forward that we believe will improve the paper. In particular, we acknowledge that not all of the issues raised in our paper are novel, and this is not what we had intended. Perhaps the main objectives of this exercise, which we carefully thought about through revisions and discussions, have been misconstrued by our choice of the title of the paper (*Biogeochemistry: Its Future Role in Interdisciplinary Frontiers*). We have now changed the title to: *Ideas and perspectives: Biogeochemistry – Some Key Foci for the Future.* The revised article now reflects what we consider some key topics in biogeochemistry that are rapidly developing, and/or in need of continued emphasis over the next decade. In general, our revisions are as follows: 1) a new title for the paper with some further clarification of our main goal; 2) a shorter and more focused document, with omission of the section of spatial/temporal advancements; and 2) more clearly emphasized the key points on new linkages with modern and past eco-evolutionary processes and continued emphasis on the social sciences (discussed below) - with additional supporting references.

**Rev. 1.**

*1. "I have mixed reaction to this paper. On the one hand, how can one criticize a plea for more interdisciplinary perspectives, especially into genomics and the social sciences? And one certainly cannot criticize pleas for better communication of our results to improve science-based policy. On the other hand, all this has been said before, widely and frequently, so I really didn't learn much from this perspectives piece, nor did it offer new excitement."*

The issues raised in this paper were not meant to be exclusively novel, but more a reminder of where biogeochemistry needs to keep moving and where enhanced development and greater efforts are needed, which some specific examples of where past efforts have not yet succeeded. The audience is also intended to be broad including not only established biogeochemists but also early career scientists new to the field who may be particularly interested in its societal relevance. As we mention in our perspective, the Biogeosciences were only marginally prepared to understand how changes in economic activity associated with COVID19 affected air and water quality and carbon fluxes. Similarly, the necessary multidisciplinary interactions are still not well represented in new funding programs in such areas as Critical Zone science and Biological Integration in the U.S. and elsewhere. These along with other key topics in our paper are reminders of what this team of biogeochemists' view as key areas and something we feel is worthy of publication.

*2. "The very name: Bio-Geo-Chemistry, reflects the interdisciplinary roots of this field of endeavor. In several papers, Paul Falkowski and Diane Newman have reviewed how the evolutionary biochemistry of prokaryotes has left its mark on the Earth's chemical conditions."*

Yes, and the paper by Falkowski, which is a key figure in a recent paper published by Bianchi (2020), on the history of biogeochemistry, along with work by Newman, are now cited as key papers that links Earths' prokaryotic history and future biogeochemistry. A key point we now emphasize in the paper, is the importance of using proxies and fossil communities to better understand not just changes in community structure, but the impact these metazoan (not just microbial) changes have had, and will have, on key biogeochemical drivers, such as redox. While range expansion has been a notable topic in recent years, much of the emphasis has been on the consequences of changing diversity and microevolution, and not how these changes affect community structure along with associated biogeochemical properties and processes, for example, redox and changes in bioturbation and/or bio-erosion. We have added some clarity to this in the paper.

*3. "Even the importance of integrating the social sciences in environmental science is widely recognized program of coupled human ˇ and natural systems at NSF and a requirement by the agency for social science linkages in its long-term ecological research programs. Better transmission of our results to policy makers was named Translational Ecology by Schlesinger in a 2010 editorial in Science and now the focus of a working group at the National Socio-Environmental Synthesis Center (SESYNC) at the University of Maryland."*

While these points about multidisciplinary interactions, especially with social science, have been made before, there are still both vast unexplored opportunities and an urgent need for more interactions, and a reassertion these issues along with other newly developing complexities are needed. For example, while translational biogeochemistry was certainly a notable and timely concept, something we clearly missed and have cited in our revisions, we have witnessed recently protests related to carbon taxes that indicate need for new approaches and integration of biogeochemistry in relation to social science, and in particular the feedbacks with human behavior.  The latter is a very embryonic line of research which we feel may be new to many in the scientific community. Once again, a reassessment and unification of ideas in our paper looking ahead for the next decade or so seems important, especially with broad the diversity of fields identify themselves under the Biogeosciences "umbrella."  We have now referenced the earlier views on *Translational Ecology* to add better historical context to what we are saying here. We have also suggested that previous lack of progress on integration between social and natural sciences has been stymied by lack of data on the coupling between the two types of systems, and we argue that digital social data may serve as a source of data to 'jump-start' closer integration of these fields.

**Commented [MOU1]:** Should this be ecology?

*4. "I have a fear that the statements about rapid evolution (page 3, line 79ff) are at odds with the recognized inefficiency of eugenics. So, I have no real criticisms of this piece; I just question its novelty and potential impact amongst the crowded pages on our computers that already sap our time and energy for forward progress."*

We acknowledge that perhaps we were not clear enough on this point, and have made the necessary revisions.

**Rev. 2.**

*1. "This paper has an excellent group of authors. However, I agree with the comments of   Bill Schlesinger – it's not clear what hasn't already been said many times before. The abstract drives that home. What is really new in this paper?"*

See our aforementioned response to Reviewer 1 on this issue of novelty.

*2. "So what might be a way forward? One suggestion is to develop a list of what are the new insights in this manuscript, and write the text around those instead of trying to cover so many different angles (scales, topics, etc).  That said, the paper is well written in that it was very easy to read. I just didn't come away with new insights."*

*These are excellent suggestions by the reviewer.* Consequently, we have now changed the title and focused on key topics with more clarity.  In particular, we have omitted the spatial-temporal advances made in sampling capabilities etc., and now focus just on the need for new biogeochemical linkages in modern and past eco-evolutionary processes and continued emphasis on social sciences – in light of global change such as human population growth and zoonotic pandemics, range expansion by organisms (including human emigration/immigration patterns), weather disasters etc.

We believe the perspective we offer now provides value for the biogeochemistry community through organizing and assessing a few key areas for progress. We intend that our essay stimulates assessment. We advocate for advances in areas we currently recognize, and we anticipate that via discussion and debate differing perspectives held by others will emerge to drive the field forward.

---

## Author Response (AR2)

Abstract, line 25-27: point (2) is no longer relevant, I feel – since this is no longer discussed in detail in the revised manuscript. While this point remains valid, I don't feel it should remain in the abstract.

L87-88: remphrase to: […] for (1) better integration of […], and (2) continued integration of […]

**Okay, now corrected**

L105: here you mention that this takes place over 'just a few generations', on L 139 this is referred to as 'just a few tens of generations'. This should be clarified/harmonized.

**Now corrected**

P4-5. IN section 2, I feel paragraph 2 (starting at L 105: Recently, evidence has shown …) connects directly to paragraph 4 (starting at L135: There is now ample evidence for rapid evolution …), while the 3rd paragraph is a bit lost in between and should be moved to towards the end of this section (below the current L150 for example). There is perhaps some duplication in par 2 and 4 that can be merged.

**Now corrected**

L 151-152: This is a bit of an awkward sentence, and grammatically incorrect (the fact that the evolution […] have occurred rarely […] ?)

**Now corrected**

Figure 2: the description of the connection between individuals and the atmospheric domain is somewhat cryptic. "chemical emissions" – can this be reformulated ?

**I am not sure why you think it is cryptic, same something more specific like contaminants or pollutants I would not apply equally/appropriately to the constituents we list there, we did talk about this when the figure was made.**

L178: "In developed countries" can be removed I think. The example cited here is from France and this is mentioned explicitly. Perhaps add a year to these events – I can assume that for newer generations this will not be familiar in the future.

**Now corrected**

L186-187: "the primary barrier […] are often political and social limitations" ▯ should be: "the primary barriers" ?

**Now corrected**

L196-197: rephrase this section, not clear (there is a verb missing I assume): " […] where evidence exists for how organismal range expansion and/or contraction of say, marine benthic communities in response to climate change."

**Now corrected**

L199: impacts: should this be 'pressures' ?

**Now corrected**

L203: "will be key in understanding and laying the foundation for good policy development." Key in understanding what ?

**Now corrected**

L204: "have long been essential in the development in part, in NASA exploration […]": there seems to be something missing there.

**Now corrected**

L205: "biogeochemistry in uniquely poised": is uniquely poised.

**Now corrected**

L 222: what do you mean here with "biogeochemical systems pathways" and how are extreme climate events and ocean acidifications examples thereof ?

**Now changed**

L237-240: A few issues with this sentence. First, "A keys to success" should be "A key to success", and the section starting with "in predicting future changes …" is awkward, not clear what that links to. Or perhaps it's just missing a comma before "in predicting".

**Now corrected**

L249-254: here too, a few awkward sections. "in an already complex mixture of interdisciplinary world in the sciences" ? Further on, "a way forward in planning for humans can better plan for their, and other organisms range expansion, in response to climate change, on the planet.". There's some editing to do there to make this sentence work, perhaps this sentence should be split to make it easier. "on the planet" can likely be removed (given that I do not see where else ?)

**Okay, now changed.**

Reference list: I did not check throughout, but I notice some references cited in the text are missing (e.g. Han et al. 2015, Allen et al. 2017, Delgado-Baquerizo et al., 2020) and that other references are incomplete (e.g. Carney & Krausse 2020). Please go through the text and ensure that all references cited are provided in the reference list, and then cross-check if all references in the list are also cited in the text in case some did not make it into the final version. Tanhua et al. (2019) in reference list contains the first author twice. There are two Saito et al. (2020) references in the list, make these 2020a and 2020b and specify in the text which is which. I only looked at a selection, so there might be more issues than those listed here.

**Now corrected and checked.**

---

## Author Response (AR3)

Associate Editor Decision: Publish subject to technical corrections (01 Apr 2021) by Steven Bouillon

Comments to the Author:

Dear Thomas and co-authors,

Thank you for the revisions.

A few technical corrections from the earlier list remain:

One point was not addressed: L237-240 (now L264-267 in the revised version): A few issues with this sentence. First, "A keys to success" should be "A key to success", and the section starting with "in predicting future changes …" is awkward, not clear what that links to. Or perhaps it's just missing a comma before "in predicting".

**Now corrected**

References: still a number issues to fix there, I went through it in both directions but please double-check before uploading your final manuscript files.

Saito et al. (2020a/2020b): this has been fixed in the references, but not in the text. Please specify on L76 and L165 which is 2020a and which is 2020b.

**Now corrected**

Collins & Bell (2006) in text, but Bell & Collins (2006) in reference list.

**Now corrected ii is supposed to be Bell and Collins 2008**

L99 and L102: Butois et al. (2020) : should be Buatois et al. (2020)

**Now corrected**

L101: Boyle et al. (2014): not in reference list.

**Now added**

L108 : Bell et al. (2008) : not in reference list.

**Now added**

L323: Crain et al. (2014) : not cited in text.

**Now added**

L340 : Derouin (2020) : not cited in text.

**Now added**

Best regards

Steven Bouillon

**Steve, thank you for your careful review of this, it really helped!**